# Evaluating the use of large language models for post optical character recognition correction in Brazilian Portuguese

## Abstract

In recent decades digital media have taken precedence over printed media, firmly establishing themselves in everyday life. Optical Character Recognition (OCR) technology facilitates the digitization of printed text but frequently introduces errors during the process. This study investigates the effectiveness of generative Large Language Models (LLMs), like the model Gemma 3, in correcting OCR outputs in Brazilian Portuguese. Using the ESTER-Pt dataset, we assess the models' ability to leverage contextual information to identify and correct OCR-induced errors. The results demonstrate that LLMs can significantly outperform existing methods, achieving an improvement in character error rate (CER) over the current state of the art in Portuguese, reducing it from 5.12 to 1.69.

## 1 Introduction

The digitization of documents is essential to better preserve cultural heritage and to enable processing by computers, for information retrieval or summarization, for example (Kanerva et al., 2025; Nguyen et al., 2021). This process, digitizing images into text files as an automated task, is called optical character recognition (OCR). According to Nguyen et al. (2021), this technique has become one of the most popular methods for digitizing printed text.

OCR is a computer vision task for extracting editable text from images, enabling the digital processing and editing of such text. This task is hard, given that images may be noisy and distorted, as well as having incorrect spelling. In historic texts there is additional complications, since they may exhibit archaic spelling, or they may be written in unusual typographic fonts (Boros et al., 2022; Bazzo et al., 2020).

Even with the current OCR engines, such as Tesseract (Smith, 2007), the results often still contain errors, such as misrecognized characters, incorrect words, or improper formatting. There are studies that analyze the characteristics of OCR errors for the English language, such as Jatowt et al. (2019). According to Bazzo et al. (2020); Boros et al. (2022) and Chiron et al. (2017b), OCR errors negatively impact natural language processing (NLP) tasks, such as information retrieval, even when these errors are minimal. In this context, the post-OCR correction task seeks to identify and fix the errors of a previously performed noisy OCR.

Post-OCR correction has been an active research area in the last decades (Nguyen et al., 2021). And, recently, large language models based on the Transformer architecture have revolutionized natural language processing.

Considering the novelty of this approach, using Transformer large language models, and the shortage of research on post-OCR correction for Brazilian Portuguese using LLMs, a significant gap on the literature is identified. Given this context, this work aims to investigate the usage of these advanced models to post-OCR correction tasks in Brazilian Portuguese, contributing to filling this scientific gap.

The objective of this work is to evaluate the performance of a post-OCR correction pipeline using large language models based on Transformers and to compare it with the performance of traditional spell-checking models. The research focuses on analyzing the effectiveness of these models in correcting

common errors found in digitized texts, such as character recognition mistakes and misinterpreted words.

## 2 RELATED WORK

On this Section, previous work related to this task is discussed. The works are arranged in three Subsections, the first has works that study OCR errors and their impact on other tasks. The second contains works that study semi-automated methods, and, finnaly, on the third section works that introduced automated methods are presented, including some works on non-English languages, with a special focus on Brazilian Portuguese works.

### 2.1 OCR ERROR ANALYSIS

To better understand OCR errors, Jatowt et al. (2019) conducted a statistical analysis of four datasets of errors produced by the OCR process, comparing them to spelling errors made by humans. The statistical analysis enabled the creation of synthetic OCR error datasets, generated by the automatic insertion of errors into correct texts, as in the work of Santos et al. (2023), which introduced a synthetic dataset in Portuguese, alongside datasets constructed using other techniques. The work of Santos et al. (2023) is especially important, as Portuguese is, according to the authors themselves, underrepresented in linguistic resources for this task. More about the datasets of Santos et al. (2023) can be found in Subsection 3.1.

Correcting these errors is critical, as noisy data affects other NLP tasks, as Bazzo et al. (2020) demonstrated in their study analyzing how OCR-induced errors impact the Information Retrieval task, showing that even a 5% error rate produces a significant impact. To organize the existing studies on this task, Nguyen et al. (2021) conducted a state-of-the-art survey on the post-OCR correction task, formally defining the problem and compiling the various techniques.

### 2.2 SEMI-AUTOMATED POST-OCR CORRECTION

According to Nguyen et al. (2021), the cost of manually digitizing a book is around one euro per page. This is a very high cost for most applications, such as digitizing a library with a huge number of volumes. To reduce this cost, Von Ahn et al. (2008) proposed using CAPTCHA tests to manualy refine OCR on words that are difficult to recognize automatically.

Another work that enables semi-automated post-OCR correction is that of Vobl et al. (2014), which presents PoCoTo, a tool with a graphical interface that displays possible OCR errors alongside the original image and allows for the quick correction of errors in digitization. Building on this work, Englmeier et al. (2019) presented two extensions to PoCoTo, A-PoCoTo and A-I-PoCoTo, which utilize the output of multiple OCR engines and add further automated steps to the process, respectively.

### 2.3 AUTOMATED POST-OCR CORRECTION

According to Nguyen et al. (2021), automated post-OCR correction methods may be divided into two categories: approaches focused on separate, or local, words, and approaches that take context into account, called Context Leveraging OCR Correction (CLOCR-C) by Bourne (2025a). Local approaches were the first to emerge, using dictionaries and probabilistic inverse OCR error models to identify and correct errors. In the years 2017 and 2019, competitions for post-OCR correction were organized by ICDAR (Chiron et al., 2017a; Rigaud et al., 2019). The best solution in the 2017 competition (Chiron et al., 2017a) employed a local approach, using machine learning and analyzing a four-character window around each character.

The 2019 competition (Rigaud et al., 2019), on the other hand, had as its best technique a BERT model (Devlin et al., 2019) fine-tuned, thus using context to correct errors. Among these approaches, it is possible to distinguish two groups: works that use in-context learning, methods that use general language models and carefully crafted task descriptions, as well as examples, and those that perform fine-tuning, an additional training step on pre-trained models with data from the specific task.

An example of work that explores fine-tuning is that of Bourne (2025b), who developed a Python library to synthetically generate errors similar to OCR, using Markov chains. Next, the author performed fine-tuning of generative Llama models (Touvron et al., 2023; Dubey et al., 2024) for Post-OCR correction, demonstrating that models adjusted with synthetic data outperform models trained only on real data in English.

Veninga (2024) performed fine-tuning of a ByT5 (Xue et al., 2021) encoder-decoder model, comparing it with a Llama model where only zero-shot and few-shot learning were used. The study demonstrated that the model adapted specifically for this task outperformed the general model.

Next, among works that use in-context learning, Bourne (2025a) conducted a study on three historical datasets in English. Investigating the performance of seven generative LLMs, including some commercial LLMs, to examine the impact of using text that underwent post-OCR correction in a Named Entity Recognition task. In his results, the author demonstrated that LLMs are capable of performing post-OCR correction, even improving performance in Named Entity Recognition tasks. He concludes that the ability of LLMs to leverage context allows them to perform post-OCR correction, even without additional training, in a form of in-context learning.

Thomas et al. (2024) compared the Llama 2 model with a fine-tuned BART and demonstrated that the general Llama model performed better than the BART model in English. Similarly, Kanerva et al. (2025) investigated the use of LLMs for post-OCR correction of historical documents in English and Finnish, demonstrating that LLMs can correct OCR outputs in English, but failed to show any positive results in Finnish. This work also presented some methods to address artifacts generated by generative LLMs.

For Portuguese, Suarez Vargas et al. (2021) presented sOCRates, a post-OCR correction method that uses a BERT model and convolutional layers to detect errors. The correction of the detected errors was carried out using a local probabilistic model, which generates a list of candidates and analyzes bigram probabilities. The results were analyzed intrinsically, by comparing them to reference texts and evaluating the impact on error rates (CER and WER) and correction accuracy; and extrinsically, by assessing the impact of post-OCR correction step on information retrieval systems. Although sOCRates demonstrates to be the best method in intrinsic evaluation, this result does not directly reflect in the extrinsic evaluation, where although sOCRates improves on the original text, the best results were using the spelling correction system, SymSpell. The authors postulate that this is due to the correction system favoring common words, which do not add relevant context for information retrieval systems.

Also working with texts in Portuguese, de Araújo et al. (2024) investigated post-OCR correction in a variant of OCR for handwritten texts, using fine-tuning of four language models, all of them adapted to Portuguese: Sabiá, adapted from Llama 1; Gervásio, adapted from Llama 2; BART Base Portuguese (BART PT), adapted from BART; and ByT5 Small Portuguese, adapted from ByT5. Of these, only ByT5 and BART PT, both with fine-tuning, produced a text with an error rate lower than that of the original text.

Based on the context that was presented, this work aims to evaluate a post-OCR correction method in Portuguese using in-context learning, as Kanerva et al. (2025) did for English and Finnish, on the dataset published by Santos et al. (2023). As the aforementioned dataset is public, it is possible to perform a comparison with other variables, including more recent techniques and large language models. Thus, this work seeks to contribute to the advancement of post-OCR correction research in Portuguese, exploring the effectiveness of Transformer-based language models, given that the current literature, to the best knowledge of the authors, is scarce in this context.

## 3 METHODOLOGY

To evaluate the performance of LLMs in the task of correcting digitized documents using OCR, tests were conducted with the ESTER-Pt dataset provided by Santos et al. (2023). A post-OCR correction pipeline was implemented, in which the text was divided into chunks and each of them was processed individually by the model, and finally the chunks were merged to generate the corrected text. Finally, the results were compared to the reference texts and evaluated using metrics from the research field. Figure 1 illustrates the proposed post-OCR correction pipeline.

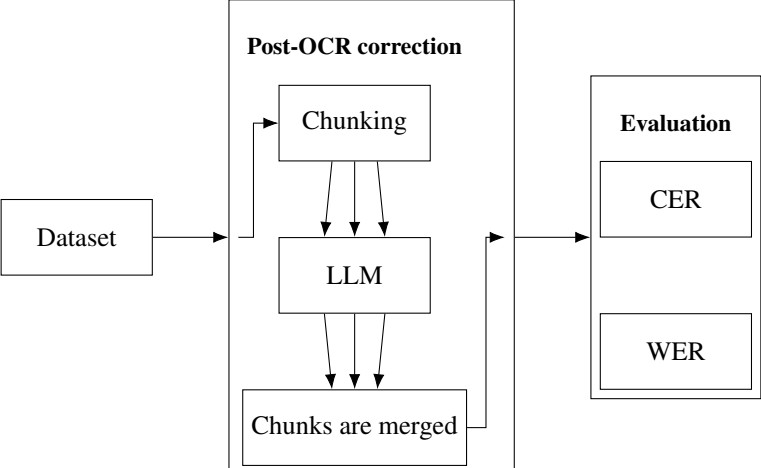

Figure 1: Post-OCR correction pipeline. The dataset is processed in chunks and each of them is corrected by a language model, the results are then merged and evaluated.

### 3.1 DATASET

The dataset ESTER-PT by Santos et al. (2023) is a dataset for Post-OCR correction in the Brazilian Portuguese language composed of four partitions with distinct characteristics, as listed below: *(i)* ESTER-PT-STB: Synthetic dataset built from Portuguese Wikipedia pages with synthetically inserted errors similar to OCR errors. The dataset consists of texts with errors and reference texts. *(ii)* ESTER-PT-SIB: Synthetic dataset built from the same Wikipedia pages as the dataset ESTER-PT-STB. These pages were formatted as images simulating scanned pages. The dataset consists of images and reference texts. *(iii)* ESTER-PT-HIB: dataset containing images of real documents from various sources with artificially added distortions based on several degradation models. It consists of images and reference texts. *(iv)* ESTER-PT-RIB: dataset consisting of images of real books and documents. The reference texts were extracted from Project Gutenberg.[1]

### 3.2 POST-OCR CORRECTION

To perform post-OCR correction, the original text is divided into chunks that were processed independently by the LLM and then regrouped. The process of chunking will be discussed in detail in Subsubsection 3.2.1. And finally, Subsubsection 3.2.2 provides details on the models and the specifics for their execution.

#### 3.2.1 CHUNKING

To avoid issues with the context window size of language models, it is necessary to split the texts into smaller segments, called chunks, with a maximum size of $C$. After conducting some preliminary experiments, the following chunking procedure was established: the original text is divided into paragraphs, and each paragraph is processed separately. Next, each paragraph is split at all periods followed by an uppercase letter, with any amount of spaces between them, and then these pieces are regrouped, attempting to maximize the size of each chunk without exceeding the maximum size $C$.

#### 3.2.2 LARGE LANGUAGE MODELS

After splitting the text into chunks, each one was processed by an LLM model that was responsible for correcting the OCR errors present in the text. Four language models were evaluated, all of them with Q4_0 quantization, as shown in Table 1.

---

[1] `https://www.gutenberg.org/`, accessed on June 23, 2025.

Table 1: Large Language Models that were evaluated.

| Model | Source |
|---|---|
| Gemma 3 12B | Gemma Team (2025) |
| Gemma 3 27B | Gemma Team (2025) |
| Llama 3.1 8B | Dubey et al. (2024) |
| Mistral v0.3 7B | Jiang et al. (2023) |

The models were run using the ollama[2] library, which is an open-source tool for running language models. For each chunk of text, the language model was executed with a prompt, as shown in Figure 2. The prompt used here emerged from various preliminary experiments that highlighted the need to emphasize to the model that it should not alter the wording of the original text, only correct character interpretation errors. The prompt used is in Brazilian Portuguese, as all the texts in the dataset are in this language. A translation of the prompt to English is also provided in Figure 2b for better understanding.

**System:** Você é um assistente de IA que corrige erros de OCR em textos. Você deve corrigir os erros de OCR no texto fornecido. Você deve manter o texto o mais próximo possível do original, mas corrigir os erros de OCR. Você deve evitar adicionar ou remover palavras desnecessárias. Você deve evitar adicionar ou remover caracteres desnecessários. Apenas retorne o texto corrigido, sem explicações ou comentários.
**User:** ( Text to be corrected )

**System:** You are an AI assistant that corrects OCR errors in texts. You must correct the OCR errors in the provided text. You must keep the text as close as possible to the original, but correct the OCR errors. You must avoid adding or removing unnecessary words. You must avoid adding or removing unnecessary characters. Just return the corrected text, without explanations or comments.
**User:** ( Text to be corrected )

(a)

(b)

Figure 2: The prompt that was used to correct the OCR errors. a) Original prompt in Brazilian Portuguese. b) Prompt translated to English for better understanding.

The model's response is then processed in order to remove generation artifacts. This way, the line that is the closest in length to the input text is extracted along with any additional spaces, which are then removed. Spaces that were around the original chunk are added, and finally, they are concatenated.

### 3.3 EVALUATION METRICS AND ENVIROMENT

To evaluate the performance of the post-OCR correction method, two automatic metrics were used: CER and WER. They are, respectively, the character error rate and word error rate. They were chosen for being widely used in post-OCR correction works, as stated by Nguyen et al. (2021). The implementation of the metrics used was from the Python package `evaluate`.[3]

The experiments in this work utilized the resources of the infrastructure of *(Removed due to double-blind review)*. They were run on machines with the following configuration: Intel(R) Core(TM) i9-14900KF CPU, NVIDIA GeForce RTX 4090 GPU, and at least 128 GB of DDR5 RAM.

---

[2]`https://ollama.com/`, accessed on June 23, 2025.
[3]`https://huggingface.co/docs/evaluate/index`, accessed on June 23, 2025

## 4 VALIDATION

In this chapter, the experiments carried out to evaluate the use of LLMs for post-OCR correction (Subsection 4.1) and the results with discussion (Subsection 4.2) are presented.

### 4.1 EXPERIMENTS

The objective of the experiments in this work is to choose the best possible parameter configuration for the use of LLMs for post-OCR correction. After determining the ideal configuration, the results will be compared with other techniques. To determine the best configuration, a random partition of the original dataset ESTER-PT-STB was created with $2\%$ of the original size by randomly sampling 100 of the $5,000$ pages. This partition will be called ESTER-PT-STB$_{Small}$ or $STB_{Small}$ and was used for the initial experiments.

The model was selected through an experiment in which all items from Table 1 were tested with the following parameters: chunk size ($C$) of $512$, this value was picked arbitrarily but proved suitable in preliminary tests; the model temperature was set to $0.26$, a value initially chosen as it was the ideal value found by Kanerva et al. (2025) for post-OCR correction in English. The models executed the pipeline described in Chapter 3 using the dataset $STB_{Small}$.

To evaluate the impact of varying the model's temperature on post-OCR correction, the pipeline was assessed at temperatures $0.14$ and $0.26$, the ideal temperatures found by Kanerva et al. (2025) for post-OCR correction in Finnish and English, respectively; and then in increments of one-tenth up to $0.9$ (0,3; 0,4; 0,5; 0,6; 0,7; 0,8; and 0,9). For this experiment the size of the chunks ($C$) was fixed on $512$. The model used was the best found among the model tests, the Gemma 3 27B.

To find the ideal chunk size for the pipeline, the following sizes were tested: 64, 128, 192, 256, 384, 512, 1024. The model and temperature used were the best from the previous tests, the Gemma 3 27B model and a temperature of $0.26$.

After finding the parameters in the previous experiments (Table 2), a test of the model was conducted with the full ESTER-PT-STB dataset to obtain comparative results with other techniques that used it. The results of the other techniques, sOCRates (Suarez Vargas et al., 2021) and SymSpell, on the ESTER-PT-STB dataset were obtained from Santos et al. (2023). It is important to emphasize that it was not possible to reproduce these techniques, as the execution of sOCRates required additional files that are no longer hosted, and SymSpell requires a frequency dictionary, which is not specified in the work for reproduction.

Table 2: Parameters for comparative testing.

| Parameter | Value |
|---|---|
| Model | Gemma 3 27B |
| Temperature | 0.26 |
| Chunk Size ($C$) | 1024 |

### 4.2 RESULTS AND DISCUSSION

This section will present the results for the four experiments followed by a brief discussion highlighting the main findings.

The results of the initial tests with each model are shown in Table 3. It can be observed that the Gemma 3 models of 12B and 27B achieved similar results, even though one of them has twice the number of parameters of the other. In this way, the 27B model still achieved a slightly better result. The Llama 3.1 8B model had a worse CER than the original text. The Mistral 7B model performed worse than the original text on both metrics.

The results of the temperature variation experiment of the Gemma 3 27B model are in Table 4. The results for temperatures between 0.14 and 0.70 are very similar, so it is difficult to conclude that any of these temperatures is better. Temperatures above 0.70 begin to show a degradation in the results.

Table 3: Results for the various models on the dataset $STB_{Small}$. Original Text refers to the raw text generated by the OCR, before any correction.

| Model | CER (%) ↓ | WER (%) ↓ |
|---|---|---|
| *Original Text* | 2.26 | 13.68 |
| Gemma 3 12B | 1.78 | **6.97** |
| **Gemma 3 27B** | **1.64** | 6.69 |
| Mistral 7B | 8.24 | 16.51 |
| Llama 3.1 8B | 3.21 | 9.62 |

↓ value should be minimized, **bold** best result for each metric.

One possible explanation is that for temperatures below 0.70 the model reaches the limit of what can be corrected with the dataset $STB_{Small}$, and that temperatures above 0.70 begin to generate artifacts that impair the correction. As there is no significantly better result, the temperature of 0.26 was chosen for the next experiments.

Table 4: Results for the variation of the model's temperature on the dataset $STB_{Small}$. Original Text refers to the raw text generated by the OCR, before any correction.

| Temperature | CER (%) ↓ | WER (%) ↓ |
|---|---|---|
| *Original Text* | 2.6 | 13.68 |
| 0.14 | **1.65** | 6.70 |
| **0.26** | **1.65** | 6.70 |
| 0.30 | **1.65** | 6.71 |
| 0.40 | **1.65** | **6.69** |
| 0.50 | **1.65** | 6.73 |
| 0.60 | 1.67 | 6.72 |
| 0.70 | 1.67 | 6.71 |
| 0.80 | 1.70 | 6.80 |
| 0.90 | 1.71 | 6.84 |

↓ value should be minimized, **bold** best result for each metric.

The results when varying the chunk size ($C$) were presented in Table 5. Chunk sizes of 192 and above show a better CER than the original text. For WER, any chunk size was sufficient to outperform the original text. The metrics tended to have better results with larger chunk sizes, with a small exception in CER for $C = 1024$, which regresses slightly. Thus, it is possible to conclude that increasing the chunk size improves post-OCR correction, as it allows the model to have more context to correct the errors.

Table 5: Results for varying the chunk size ($C$) on the dataset $STB_{Small}$. Original Text refers to the raw text generated by the OCR, before any correction.

| $C$ | CER (%) ↓ | WER (%) ↓ |
|---|---|---|
| *Original Text* | 2.26 | 13.68 |
| 64 | 2.87 | 11.08 |
| 128 | 2.34 | 9.60 |
| 192 | 2.11 | 8.34 |
| 256 | 1.85 | 7.53 |
| 384 | 1.67 | 6.86 |
| 512 | **1.66** | 6.73 |
| **1024** | 1.71 | **6.44** |

↓ value should be minimized, **bold** best result for each metric.

The post-OCR correction results of the complete dataset ESTER-PT-STB with the parameter configuration found in this work are in Table 6 along with the results of other techniques for the same dataset.

Thus, it is possible to observe that the post-OCR correction method presented in this work outperforms the other methods in CER and WER. This demonstrates that LLMs can surpass post-OCR correction techniques by using context to infer the correct text.

Table 6: Comparative results of LLMs with other post-OCR correction techniques, using the complete dataset ESTER-PT-STB. Original Text refers to the raw text generated by the OCR, before any correction.

| Technique | CER (%) ↓ | WER (%) ↓ |
|---|---|---|
| *Original Text* | 2.23 | 13.39 |
| *Ours* | **1.69** | **6.22** |
| sOCRates[*] | 5.12[*] | 11.97[*] |
| Symspell[*] | 5.61[*] | 17.07[*] |

[*]Results obtained from Santos et al. (2023),
↓ value should be minimized, **bold** best result for each metric.

## 5 CONCLUSION

In this work, a post-OCR correction pipeline in Brazilian Portuguese using LLMs was presented. It was evaluated using various configurations and compared with other state-of-the-art techniques. The work consisted of evaluating the performance of the post-OCR correction pipeline using language models based on the Transformer architecture and comparing the performance with traditional models.

Results show that, with the use of LLMs, we were able to improve the state of the art from 5.12 to 1.69 in CER, and from 11.97 to 6.22 in WER. These results demonstrate that LLMs can be a useful tool in building automated post-OCR correction techniques in Portuguese, surpassing the traditional methods found in the literature.

But despite the contributions that were highlighted, this work has some shortcomings: for the choice of the parameters, because a greedy approach was used, other models were very quickly eliminated, even when they could have performed better with more suitable parameters. Also, among the datasets provided by Santos et al. (2023) only the synthetic dataset was used. This dataset behaves better and has the lowest CER. It is not possible to state whether, with noisier data, the results would be different. The dataset ESTER-PT-STB used for the tests was compiled using Portuguese Wikipedia pages; it is not possible to assert whether the evaluated models had not been previously trained on these Wikipedia data, which would constitute data leakage.

For future work in post-OCR correction using language models, some areas left for improvement by this work are suggested here. An interesting possibility is to evaluate various prompts, analyzing how they affect the quality of the generation. Another important point involves the potential creation of datasets for this task in Portuguese, as the only one found was that of Santos et al. (2023), limiting the experiment. Furthermore, the use of other non-synthetic datasets may prove interesting to evaluate behavior in scenarios closer to real-world conditions.

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

## A  LLM USAGE

LLMs were used sparingly for polishing the writing and reviewing the text.

