# OpenReview forum: "Evaluating the use of large language models for post optical character recognition correction in Brazilian Portuguese"
_ICLR.cc/2026/Conference — Submitted to ICLR 2026_

### Official Review · Reviewer_9f5i · 2025-10-22

**Soundness:** 2
**Presentation:** 2
**Contribution:** 2
**Rating:** 2
**Confidence:** 5

**Summary:**

This paper evaluates the effectiveness of using generative Large Language Models (LLMs), specifically models like Gemma, Llama 3.1, and Mistral, for post-OCR correction in Brazilian Portuguese. The authors employ a zero-shot, in-context learning approach, where the OCR-processed text is split into chunks and fed to the LLM with a carefully crafted prompt instructing it to correct errors while preserving the original content. The study uses the synthetic portion of the ESTER-Pt dataset for evaluation and systematically tests different models and hyperparameters (chunk size, temperature). The results are compelling, showing that the Gemma 3 27B model significantly improves the state-of-the-art, reducing the Character Error Rate (CER) from 5.12% to 1.69%.

**Strengths:**

The authors methodically evaluate several recent LLMs and explore the impact of key hyperparameters like temperature and chunk size, providing a clear and reproducible methodology for achieving their results.

**Weaknesses:**

1. The experiments are confined to a synthetic dataset where errors are artificially inserted. The paper lacks evaluation on the non-synthetic, real-world document portions of the ESTER-Pt dataset, making it unclear how the method would perform on documents with more complex and authentic noise patterns.

2. The study only investigates a zero-shot prompting method. It does not compare this against a fine-tuning approach, which has been shown in other works to be highly effective for this task and would have provided a more comprehensive evaluation of LLM capabilities.

3. I do not see clear advantages of this paper over other published post-OCR approaches, and it also lacks citations to recent post-OCR literature.

- A Proposal of Post-OCR Spelling Correction Using Monolingual Byte-level Language Models
- Leveraging LLMs for Post-OCR Correction of Historical Newspapers
- Post-OCR Correction with OpenAI's GPT Models on Challenging English Prosody Texts
- Advancing Post-OCR Correction: A Comparative Study of Synthetic Data
- Neural Machine Translation with BERT for Post-OCR Error Detection and Correction
- Deep Statistical Analysis of OCR Errors for Effective Post-OCR Processing
- Effective Synthetic Data and Test-Time Adaptation for OCR Correction
- A Benchmark and Dataset for Post-OCR text correction in Sanskrit
- Post-OCR Document Correction with Large Ensembles of Character Sequence-to-Sequence Models

**Questions:**

You mention encoder–decoder models (e.g., ByT5) in the related work. Why not fine-tune them and compare them against decoder-only LLMs?

---

### Official Review · Reviewer_WtTN · 2025-10-31

**Soundness:** 2
**Presentation:** 2
**Contribution:** 2
**Rating:** 2
**Confidence:** 5

**Summary:**

This paper evaluates LLM for post-OCR correction, specifically for Brazilian Portuguese.  Results show better performance using an LLM versus other tested alternatives.

**Strengths:**

Results show improvement on the OCR task using the LLM approach.  The paper's study has value in testing methods on a non-English under-represented language.

**Weaknesses:**

Contribution/technical novelty is minor since the paper uses LLMs in a straightforward manner to correct OCR results - giving the OCR transcription in a prompt along with instructions to correct.

Only one dataset is used for evaluation, and the dataset is a synthetic dataset (synthetically inserted OCR errors), so the results may not generalize to true noisy OCR.

**Questions:**

n/a

---

### Official Review · Reviewer_1P8G · 2025-10-31

**Soundness:** 2
**Presentation:** 3
**Contribution:** 2
**Rating:** 2
**Confidence:** 5

**Summary:**

The authors present a simple but seems effective use of instruction-tuned LLMs to correct OCR output and improve recognition performance. The use case is Brazilian Portuguese texts. They used instruction-tuned LLMs, in major part the Gemma 3 27B for post-OCR correction. The pipeline is simple, a sentence-like chunking was performed and a “do not paraphrase, only fix OCR” prompt was used, chunkwise inference, and merging results.

The approach performance was evaluated via CER/WER on the ESTER-PT synthetic text split. Some ablations over temperature and chunk size are presented, and the best configuration reports gains (CER 1.69, WER 6.22) versus numbers for sOCRates and SymSpell reported in prior work.

The experimental design is not robust and has several problems with validity, for example, dataset realism, comparability to prior work, decoding/heuristic effects, and reproducibility, and that not help the claim of state of the art.

**Strengths:**

I liked the language setting (pt-BR) and the minimal prompting recipe of the proposal. The problem is important and besides the high digitalization of society today it still exists. The gains of the proposed approach are shown and the scientific significance will hinge on demonstrating robustness on image-originated noise, fair and reproduced baselines. I was expecting experiments with some fine-tuned Portuguese models.

The proposed exploration of post-OCR correction for pt-BR with general-purpose LLMs is interesting, but it seems prior work has already established LLMs as strong post-OCR correctors in English and other languages, which puts the novelty here mainly in the language and a specific minimal prompting recipe.

**Weaknesses:**

The authors do not reproduce baselines, evaluate only the synthetic split (omitting image-derived SIB/HIB/RIB), and use under-specified post-generation cleanup heuristics, which together make the state-of-the-art claim premature despite promising practical results.

The external validation is limited because the evaluation is only in the synthetic text split. For example, the image-derived splits (SIB/HIB/RIB), where OCR noise is more realistic, are not considered.

The SOTA comparison copies numbers from Santos et al. (2023) instead of reproducing with aligned preprocessing and scripts.

The “Original Text” CER mismatch across tables suggests preprocessing or reporting inconsistencies. The post-generation cleanup (“choose the line closest in length and strip spaces”) is unusual, under-specified, and may bias CER/WER.

Decoding controls are narrow (temperature only), and quantization effects are not assessed.

The authors do not present statistical significance tests, error taxonomy, or cost/latency analyses.

The Gemma-27B only slightly improves over 12B despite far higher compute.

Please verify the inconsistency of baseline CER across tables (2.26 vs 2.6).

**Questions:**

Please provide exact pseudo-code for the post-processing/cleanup. My point is, how much does it change CER/WER vs a no-cleanup baseline?

What text normalization/tokenization settings are used for CER/WER (case, punctuation, diacritics)? Do they match Santos et al. exactly?

Which exact Ollama model tags and quantization settings were used?

Can you report results on SIB, HIB, and RIB (even subsets) to assess robustness to image-originated noise and layout artifacts?

Please add an error analysis by category (diacritics, hyphenation, l/1/I, o/0, punctuation, word split/merge) and qualitative failure cases.

---

### Official Review · Reviewer_jg7g · 2025-11-01

**Soundness:** 1
**Presentation:** 2
**Contribution:** 1
**Rating:** 0
**Confidence:** 4

**Summary:**

This paper investigates the capability of generative LLMs, particularly Gemma-3, to correct OCR errors in Brazilian Portuguese text. Using the ESTER-Pt dataset, the authors evaluate whether LLMs can leverage contextual semantics to improve OCR accuracy relative to prior systems. Experimental results indicate a performance gain, reducing character error rate from 5.12 to 1.69 on the chosen benchmark.

**Strengths:**

1. OCR error correction remains a relevant practical application, especially in digitization pipelines for historical and printed text collections.

2. The study shows LLMs can significantly reduce CER compared to existing baseline methods on Portuguese OCR correction tasks.

3. The focus on Brazilian Portuguese contributes to broadening language coverage in OCR-related research.

**Weaknesses:**

1. The paper does not introduce a new dataset, method, or problem formulation. The contribution mainly lies in empirically validating that LLMs can improve OCR correction, which is an expected finding given current literature on LLM-assisted OCR post-editing.

2. The manuscript does not clearly articulate what makes Brazilian Portuguese OCR correction unique. Since Portuguese is a well-represented language in modern LLMs, the significance of focusing on Brazilian Portuguese specifically needs clearer justification.

3. The comparison with domain or task-specific OCR post-correction systems is extremely limited. The only OCR-focused baseline appears to be from 2021, and the rest are generic LLMs. While comparing with strong general LLMs is useful, the lack of competitive and recent domain-adapted baselines makes it difficult to conclude that the proposed approach is meaningfully superior in practical OCR pipelines. Including more representative OCR post-correction models (e.g., seq-to-seq correction models, transformer-based correction pipelines, or multilingual OCR-specialized systems) would greatly strengthen the validity of the findings.

4. The paper lacks sufficient methodological detail: metric definitions are incomplete (e.g., CER formulation), no ablations (prompting strategies, model sizes, types of OCR noise), limited dataset description, and no discussion of potential LLM pretraining contamination.

5. There is no qualitative error analysis or visual evidence (e.g., heatmaps, confusion patterns, representative correction examples). This prevents deeper insight into failure modes, linguistic challenges, or specific strengths of the LLM approach.

6. Results are shown only on one dataset, with no domain transfer tests or real-world scans. As a result, the work feels closer to a preliminary application study rather than a full research contribution aligned with ICLR standards.

**Questions:**

1. What is the novelty beyond demonstrating LLM capability? Please clarify how the work advances knowledge beyond validating a known capability.

2. Why Brazilian Portuguese specifically? Is there evidence that OCR error distributions differ significantly from European Portuguese or multilingual settings? Any linguistic analysis?

3. How is CER computed?

4. What preprocessing steps were applied?

5. Any train–test leakage concerns given LLM pretraining?

6. Why only compare to a limited set of prior model? Are classical post-editing or fine-tuning methods included?

7. Does performance hold for historical documents, noisy scans, or domain-specific corpora? Are results robust across prompts?

8. Please provide concrete example corrections and error categories (diacritics, segmentation, morphological errors, …).

---

### Meta-Review · Area_Chair_HLys · 2025-12-21

**Summary:**

All the reviewers gave a clear rejection and the authors didn't submit any rebuttal. The paper is a simple usage of LLM to correct OCR results in a zero-shot way, which is limited in novelty and contribution. I agree with the reviewers and decide to reject the paper.

**Reviewer Concerns:**

Most concerns are about the novelty of the method and the contribution of the synthetic  dataset, for which the authors provided no explanation. In an objective viewpoint, I agree that the paper is of limited value for the community.

**Reviewer Scores:**

The reviewers are expected to be maintain the decision of rejection considering no rebuttal is offered.

---

### Decision · Program_Chairs · 2026-01-26

Reject